# MoleSG: A Multi-Modality Molecular Pre-training Framework by Joint Non-overlapping Masked Reconstruction of SMILES and Graph

## Abstract

Self-supervised pre-training plays an important role in molecular representation learning because labeled molecular data are usually limited in many tasks, such as chemical property prediction and virtual screening. However, most existing molecular pre-training methods focus on one modality of molecular data, and the complementary information of two important modalities, SMILES and graph, are not fully explored. In this study, we propose a straightforward yet effective multi-modality pre-training framework for **Mole**cular **S**MILES and **G**raph (MoleSG). Specifically, the SMILES sequence data and graph data are first tokenized so that they can be processed by a unified transformer-based backbone network, which is trained by a masked reconstruction strategy. In addition, we introduce a specialized non-overlapping masking strategy to encourage fine-grained interaction between these two modalities. Experimental results show that our framework achieves state-of-the-art performance in a series of molecular property prediction tasks, and detailed ablation study demonstrates efficacy of the multi-modality structure and the masking strategy.

## 1 Introduction

Efficient molecular representation learning is foundational to drug discovery (David et al., 2020; Huang & Von Lilienfeld, 2016). With the advancement of deep learning, data-driven molecular representation learning has found applications in various domains, such as chemical property prediction (Duvenaud et al., 2015), virtual screening (Stumpfe & Bajorath, 2020), molecular design (Magar et al., 2021), and more. However, since most molecular label data need to be obtained through labor-intensive and costly wet experiments (Brown et al., 2019), there is a lack of sufficient labeled molecular data, which hinders the development of deep learning methods and can lead to issues like overfitting and poor generalization (Rong et al., 2020). Self-supervised learning holds substantial research value in addressing these challenges, which involves pre-training on unlabeled data and fine-tuning with labeled data on downstream tasks. It has shown significant promise in enhancing the performance of molecular representation learning on many downstream tasks (Xie et al., 2022).

Molecules can be described using various modalities, such as fingerprints, sequences, graphs, and more (Xia et al., 2023). Currently, molecular pre-training predominantly focuses on a single modality (Xia et al., 2023), with only a little attention given to methods jointly dealing with multiple modalities (Liu et al., 2021; Zhu et al., 2021). This paper addresses the issue of jointly pre-training on two molecule modalities: Simplified Molecular-Input Line-Entry system (SMILES) (Weininger, 1988) and molecular graph. As depicted in Figure 1, the same molecule can be represented using both a SMILES sequence and a graph, with each modality having its unique advantages and disadvantages. SMILES is a compact **implicit** representation of the molecule that excludes single-bond representation, making it well-suited for rapid compound retrieval and identification (Quirós et al., 2018). Additionally, the SMILES sequence, being a text string, can be processed with transformer-based networks well-developed in the Natural Language Processing (NLP) field for feature extraction, in which the self-attention mechanism weights and combines information from any position in the input sequence, thereby facilitating the capture of **global** contextual information (Chithrananda et al., 2020; Wang et al., 2019). However, SMILES representations only capture the relationships be-

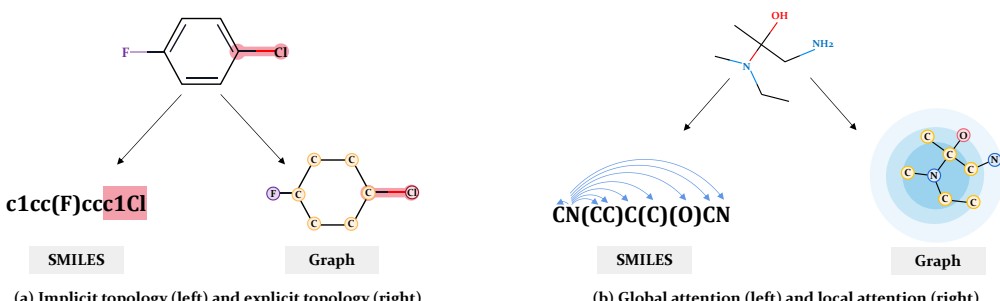

(a) Implicit topology (left) and explicit topology (right)          (b) Global attention (left) and local attention (right)

Figure 1: Comparison of two molecular representation modalities, SMILES and graph. (a) Illustration of the topological differences between the two modalities. SMILES represents topology implicitly, while graph displays explicit topology. (b) Difference in attention mechanisms used for feature processing in the two modalities. Global attention mechanism is usually used for SMILES while local attention mechanism can be easily implemented for graph.

tween atoms and bonds. They often struggle to capture the complex structural and topological information of molecules, such as the number and positions of rings, the length of side chains, and other intricate details that can be crucial in drug efficacy prediction (Lim et al., 2021; Zhang et al., 2022). Graph representations offer **explicit** portrayals of atoms, bonds, and their interconnections, showcasing the topological structure of molecules (Xiong et al., 2019). They provide detailed chemical information about molecules, including attributes for each atom such as element type, charge state, stereochemistry, and attributes for each bond, like bond type and bond length (Hall et al., 1991). However, Graph Neural Networks (GNNs), commonly used to extract features from graphs, primarily rely on message-passing layers to gather information from neighboring nodes, emphasizing the capture of **local** contextual information. This can lead to a disadvantage in capturing global context information due to information decay when delivering messages between non-adjacent nodes (Zhou et al., 2020). As a result, for the same molecule, SMILES and graph encode molecular features from different perspectives, offering complementary information. The rational combination of these two modalities holds promise for enhancing molecular representation performance.

There are several existing works on multi-modality molecular pre-training (Liu et al., 2021; Zhu et al., 2021; Liu et al., 2022). For example, GraphMVP (Liu et al., 2021) focuses on joint pre-training with 2D graphs and 3D graphs. However, these two modalities exhibit high similarity. Additionally, this study only proved 3D geometry complements 2D topology in downstream tasks, without proving 2D topology complements 3D geometry. DVMP (Zhu et al., 2021) first extracts features from SMILES and graph of the same molecule for contrastive learning. All these existing methods lack fine-grained cross-modality interactions, and there is no existing work that effectively explores the complementary information between SMILES and graph. The challenge of more efficiently combining these two modalities with significant differences lies in how to promote information exchange in fine-grain such as at the atom level rather than only achieving contrastive learning at the entire molecule level.

In this paper, we propose MoleSG, a simple yet effective pre-training framework for effectively exploring the complementary information between SMILES and graph in molecular pre-training. Specifically, recognizing that both words in SMILES sequences and graph nodes can be treated as transformer tokens (Hu et al., 2023; Huang et al., 2022), we first introduce a transformer-based unified backbone network for jointly processing embeddings from both modalities to facilitate interactions between them. Our framework consists of two independent encoders to separately convert masked SMILES and masked graph of an input molecule into token embeddings. The embeddings from the two modalities are concatenated and inputted into a standard transformer for joint processing and the output is used to reconstruct the original SMILES and graph by two specific decoders. Our framework is trained by reconstruction losses. Furthermore, to enhance cross-modality interaction, we introduce a dedicated non-overlapping masking strategy, in which we establish the positional correspondence between the SMILES sequence and the graph of a molecule to ensure that regions masked in SMILES and graph do not overlap. Intuitively, the information used for reconstructing the masked tokens can come from the context within the same modality, as well as information from the tokens of corresponding structures in the other modality. Therefore, our non-

overlapping masking strategy masks information within its own modality to encourage the model to learn information from the other modality, thereby strengthening interactions between the two modalities. To evaluate the effectiveness of MoleSG, we conduct experiments on 14 downstream tasks related to molecular property prediction and MoleSG achieves state-of-the-art (SOTA) performance in all tasks. We also compare it with the same network pre-trained by a single modality, and the experimental results show that multi-modality training learns richer molecular representation knowledge.

Our contributions are as follows: (1) We propose MoleSG, a novel molecular pre-training framework that utilizes the complementary information of SMILES and graph representations, resulting in improved performance; (2) We introduce an innovative non-overlapping masking strategy and a unified network for handling two distinct modalities, allowing for fine-grained interaction between SMILES and graph representations and achieving better representation learning; (3) MoleSG achieves SOTA performance in a series of molecular property prediction tasks, and detailed ablation study demonstrates efficacy of the multi-modality structure and the masking strategy.

## 2 RELATED WORK

**Molecular single-modality self-supervised learning:** Molecular single-modality self-supervised learning can be broadly categorized into contrastive and generative approaches. Most contrastive methods work on the modality of graph by bringing augmented graphs from the same molecule closer while pushing those from different molecules farther apart, and they focus on the global molecular information. For instance, MolCLR (Wang et al., 2022) employs diverse graph augmentation techniques for contrastive learning pre-training. FraSICL (Zhang et al., 2023) divides the same molecule into different fragment pairs based on semantics, enabling contrastive learning. KANO (Fang et al., 2023) incorporates an additional knowledge graph-based augmentation to improve the performance of contrastive learning. Generative approaches primarily predict masked molecular components using an encoder-decoder pattern, with an emphasis on learning information at the local level. For example, GROVER (Rong et al., 2020) is designed for the 2D graph modality and encompasses masked generative self-supervised tasks at the node and edge levels. Uni-mol (Zhou et al., 2023) focuses on the 3D graph modality and achieves effective 3D spatial representation learning through 3D position recovery and masked atom prediction tasks on a large dataset. Both SMILES-BERT (Wang et al., 2019) and ChemBERTa (Chithrananda et al., 2020) are designed for the SMILES modality and utilize a "cloze-style" generative pre-training approach.

**Molecular multi-modality self-supervised learning:** GraphMVP (Liu et al., 2021) leverages correspondences and consistencies between 2D graph and 3D graph to perform both contrastive and generative self-supervised learning and inject 3D information into 2D molecular graph encoders. MoleculeSTM (Liu et al., 2022) focuses on molecular graphs and text descriptions, using a contrastive learning strategy to learn the consistency between the chemical structure of molecules and their textual descriptions. DVMP (Zhu et al., 2021) addresses both SMILES and graph modalities, employing a contrastive learning approach to learn SMILES information encoded by transformer and graph information encoded by GNN from the same molecule. DVMP focuses on the same two modalities as we do but it neglects interactions between fine-grained information across different modalities.

## 3 METHOD

In this section, we will begin with providing an overview of our pre-training framework. Next, we will detail our data preprocessing procedures and introduce our innovative non-overlapping masking alignment strategy, which aims to encourage interaction between the two modalities. Following that, we will describe our network containing specialized encoders, backbone, and specialized decoders.

### 3.1 OVERVIEW OF MOLESG

As shown in Figure 2, MoleSG learns features jointly from SMILES and graph by performing masked reconstruction on both modalities with a unified feature extraction backbone network. Concretely, for a given molecule, we first convert its SMILES sequence into tokens and calculate features for nodes and edges in the graph. Then, we randomly mask some node features in the graph and then mask a portion of SMILES tokens corresponding to the remaining unmasked atoms in the graph, so that we can perform non-overlapping masking to facilitate the interaction of information between the two modalities.

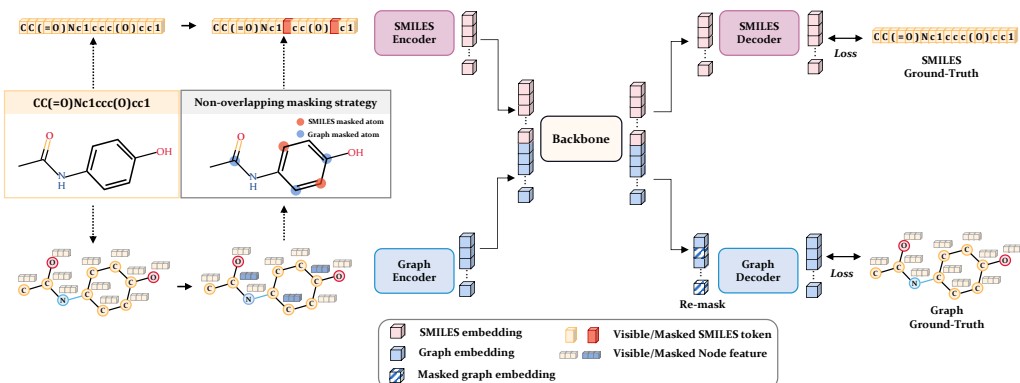

Figure 2: Overview of MoleSG. The SMILES sequence and the graph of a molecule are first randomly masked using the non-overlapping masking strategy. Then they are individually encoded by independent encoders, and the SMILES embeddings and the graph embeddings are concatenated and inputted into a transformer backbone for joint processing. Finally, processed features belonging to each modality are decoded into token ids and graph nodes for the reconstruction proxy task.

During pre-training, we employ a symmetric joint encoder-decoder framework to perform further feature extraction. The framework consists of two independent branches for the two modalities and a shared backbone for feature fusion. The independent encoder branches encode the data of two different modalities into a unified form i.e. embedding, which is suitable for understanding by a transformer backbone (Hu et al., 2023; Huang et al., 2022). The shared transformer backbone can learn the dependencies between atoms within and across the modalities and output features for the subsequent independent decoders. Finally, the SMILES decoder and the graph decoder reconstruct the original SMILES sequence and graph based on the output of the backbone.

Different from prior works (Liu et al., 2021; Zhu et al., 2021; Zhang et al., 2023), the core of MoleSG lies in the specially designed masking strategy and the unified network capable of handling data of different modalities. We will introduce the details of our masking strategy in section 3.2, followed by a comprehensive presentation of our network architectures in section 3.3-3.5.

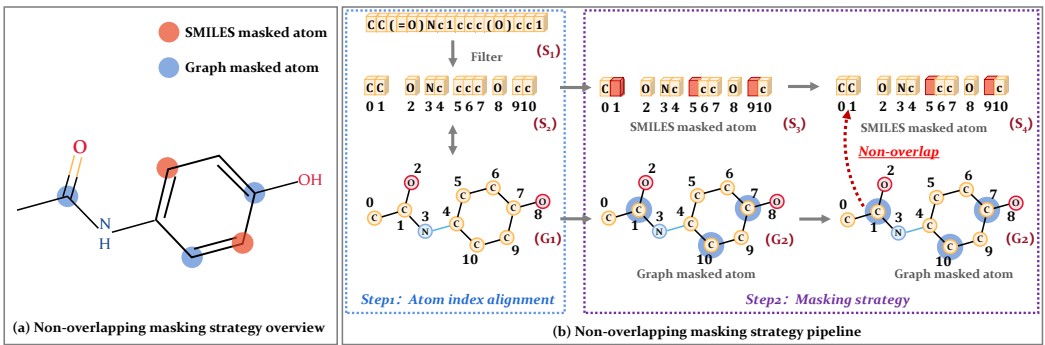

Figure 3: Non-overlapping masking strategy. (a) Non-overlapping masking strategy: Masks in the SMILES sequence and the graph for the same molecule do not overlap. (b) Non-overlapping masking strategy pipeline: First, we establish a correspondence between atom index in both modalities. Then, random masking is applied to the graph, followed by mapping the masked atoms from the graph to the SMILES sequence. Finally, random masking on the SMILES sequence is implemented on the remaining unmasked atoms of the graph.

## 3.2 NON-OVERLAPPING MASKING STRATEGY

The non-overlapping masking strategy we propose is illustrated in Figure 3, which can be divided into two steps, first performing atom index alignment between the two modalities, and then performing non-overlapping masking.

**Step 1: Atom index alignment.** Initially, for a given input molecule, we define its molecular graph as $G = (V, E)$, where $V$ and $E$ represent the sets of atoms and edges, respectively. Following the method of CoMPT (Chen et al., 2021), we precompute the node features $V_{feature} = \{v_{f0}, v_{f1}, ..., v_{f(m-1)}\}$, where $m$ is the number of atoms and then represent the SMILES sequence as the set of a series of tokens $S_1 = \{s_0, s_1, ..., s_{n-1}\}$, where $n$ is the total number of tokens. The SMILES tokens can be categorized into three classes: (1) Atoms, including single-character atoms like C and N, as well as multi-character atoms like Ca and Au, and ions like [Cl-] and [Fe+3]; (2) Chemical bonds, represented by symbols like '#' and '='; (3) Other symbols, such as numbers '1' and '2' indicating the positions of atoms in a ring and parentheses '(' and ')' denoting containing side chains. Given that single bonds are often omitted in SMILES, achieving a one-to-one correspondence between two modalities for chemical bonds is not practical. Therefore, in this paper, we focus on aligning the atom index. Therefore, we gather the tokens representing the atoms and assign indexes to them to establish a consistent correspondence between atoms in graph $G_1$ and those in filtered SMILES tokens $S_2$.

**Step 2: Masking strategy.** We randomly mask atomic features on the graph $M_G : G_1 \mapsto G_2$, where $G_2$ is the masked graph, and the set of masked atom indexes on $G_2$ is defined as $I_G$. Following that, we randomly mask atomic tokens on the SMILES sequence $M_S : S_2 \mapsto S_3$, where $S_3$ is the preliminary masked SMILES sequence, and the set of masked atom indexes on $S_3$ is denoted as $I_S$. To encourage better interaction between the two modalities, we set the overlap ratio between masked atoms in both modalities to be 0, forcing one modality to learn the "correct answer" from the other modality. Specifically, based on the one-to-one correspondence of atom index, we localize the positions of masked atoms onto the SMILES sequence. Through operation $P : I_S - I_G \cap I_S, S_3 \mapsto S_4$, where $S_4$ is the final masked SMILES sequence, we avoid masking atoms on the SMILES sequence that are already masked on the graph.

### 3.3 ENCODER

To facilitate the interaction of fine-grained features across different modalities, we use two independent encoders to convert the data of two entirely different modalities into embeddings of the same dimensions for being further processed by transformer.

For the SMILES sequence, we adopt the method used in Roberta (Liu et al., 2019b). We first convert the masked SMILES sequence into a sequence of token ids following ChemBERTa (Chithrananda et al., 2020), and we expand its vocabulary by conducting a comprehensive analysis of all tokens in our dataset, as detailed in Appendix E. Then, we calculate their corresponding embeddings $F_S \in \mathbb{R}^{N_S \times d}$ by a vanilla transformer, where $N_S$ represents the number of SMILES tokens, and $d$ is the feature dimension.

For the graph, we precompute the same node features and edge features as CoMPT (Chen et al., 2021) does. After that, a portion of node features are randomly masked, and then we feed them into the graph encoder. Our graph encoder is the same as that used in CoMPT (Chen et al., 2021), which consists of many message-passing layers. After repeating message-passing in the graph encoder, we finally obtain token embeddings $F_G \in \mathbb{R}^{N_G \times d}$ for nodes, where $N_G$ is the number of atoms, and $d$ is the feature dimension.

### 3.4 UNIFIED BACKBONE

Given that two modalities are treated as embeddings of the same dimension, we can easily use a simple unified network to learn fine-grained features in both modalities. We first add trainable parameters to $F_S \in \mathbb{R}^{N_S \times d}$ and $F_G \in \mathbb{R}^{N_G \times d}$ and then concatenate them. The concatenated embeddings $F_{S,G} \in \mathbb{R}^{(N_S+N_G) \times d}$ are then fed into the backbone. Here, we use the transformer encoder employed in Roberta (Liu et al., 2019b) as the backbone network, and its multi-head self-attention mechanism can facilitate information interaction between token embeddings both within the same modalities and across different modalities.

### 3.5 DECODER

After feature extraction in the backbone, we split the output features $F'_{S,G} \in \mathbb{R}^{(N_S+N_G) \times d}$ into features $F'_S \in \mathbb{R}^{N_S \times d}$ for SMILES and features $F'_G \in \mathbb{R}^{N_G \times d}$ for graph. $F'_S$ and $F'_G$ are features

for individual modality-specific mask reconstruction tasks. Specifically, $F'_S$ is fed into LMhead in Roberta (Liu et al., 2019b) to predict the masked token ids, while $F'_G$ is inputted into a lightweight network GIN (Xu et al., 2018) after re-masking (Hou et al., 2022) to reconstruct the masked node features. We calculate the entropy loss $\mathcal{L}_{EN}$ (Liu et al., 2019b) in SMILES reconstruction and the SCE loss $\mathcal{L}_{SCE}$ (Hou et al., 2022) in graph reconstruction. Finally, the overall loss for the entire task is as follows: $\mathcal{L}_{Total} = \mathcal{L}_{EN} + \mathcal{L}_{SCE}$.

### 3.6 FINE-TUNING

We conduct fine-tuning on 14 downstream tasks of predicting molecular properties. Since previous works only utilize a single modality in the downstream tasks, we also take a single modality as input to achieve a fair comparison. Moreover, as single modality input has an inconsistent distribution with two modalities, the backbone that takes two modalities as input during pre-training may suffer from performance decrease during fine-tuning. Therefore, we also discard the backbone during fine-tuning and inference. In other words, we only reserve a single special encoder during fine-tuning and inference. Our following experiment in section 4.3.3 also verifies it.

## 4 EXPERIMENTS

### 4.1 IMPLEMENTATION DETAILS

**Datasets setup:** During the pre-training stage, we sample 250,000 unlabeled molecules from ZINC15 (Sterling & Irwin, 2015), which is a comprehensive collection of chemical compounds for drug discovery and computational chemistry research. During the fine-tuning stage, we utilize 14 benchmark datasets from MoleculeNet (Wu et al., 2018), covering molecular data from various domains, including pharmaceuticals, biology, chemistry, and physics. These downstream datasets include 678 binary classification tasks and 19 regression tasks. For more detailed information about benchmark datasets, please refer to Appendix A.

We partition each benchmark dataset into the train, validation, and test sets in an 8:1:1 ratio. For all datasets except QM9, we employ scaffold splitting, reporting the mean and standard deviation of results from three random seeds for each benchmark. Scaffold splitting is a more challenging and realistic data partitioning method (Ramsundar et al., 2019). For the QM9 dataset, we follow the approach used in most prior work (Wang et al., 2022; Fang et al., 2023) for random splitting.

**Pre-training:** We train MoleSG for 90k iterations using the AdamW optimizer with a base learning ratio of 1e-3. We set the masking ratio for graph at 25% and for SMILES at 15%. The details of the mask ratio setting experiments for the two modes are shown in Appendix C.

**Downstream:** We set a maximum of 150 training epochs, with early stopping applied when the validation set's best value is not improved for more than 20 epochs. We use the AdamW optimizer with a base learning rate of 1e-3 and a warmup factor of 0.1 for the first 30 epochs.

**Competitors:** We compare MoleSG with both supervised (training from scratch) baselines and pre-trained baselines. Supervised methods include MPNN (Gilmer et al., 2017), DMPNN (Yang et al., 2019), CMPNN (Song et al., 2020), and CoMPT (Chen et al., 2021). Pre-training methods include N-gram (Liu et al., 2019a), PretrainGNN (Hu et al., 2019), MGSSL (Zhang et al., 2021), GROVER (Rong et al., 2020), GraphMVP (Liu et al., 2021), MolCLR (Wang et al., 2022), GEM (Fang et al., 2022), DVMP (Zhu et al., 2021), KANO (Fang et al., 2023), and Uni-mol (Zhou et al., 2023). The specific configurations for these competitors can be found in Appendix B. Additionally, for a fair comparison, we implement new MolCLR and DVMP by replacing the original encoders in them with the same networks we use, which are denoted as MolCLR$_{CoMPT}$ and DVMP$_{MoleSG}$. We also utilize our non-overlapping masking strategy in DVMP$_{MoleSG}$.

### 4.2 RESULTS OF MOLECULAR PROPERTY PREDICTION

Table 1 presents the test results in classification tasks. It can be observed that MoleSG consistently outperforms other methods across all eight datasets, demonstrating its effectiveness. It's worth noticing that though the Toxcast dataset benchmark with 617 binary classification tasks is challenging, our method still performs better than the current SOTA method KANO. Complementary information

Table 1: Performance of different models on eight classification benchmarks in physiology and biophysics. The mean and standard deviation of ROC-AUC (%) from three independent runs are reported. (Higher values indicate better performance.)

| Category | Physiology | | | | | Biophysics | | |
|---|---|---|---|---|---|---|---|---|
| Dataset | BBBP | Tox21 | ToxCast | SIDER | ClinTox | BACE | MUV | HIV |
| Molecules | 2039 | 7831 | 8575 | 1427 | 1478 | 1513 | 93807 | 41127 |
| Tasks | 1 | 12 | 617 | 27 | 2 | 1 | 17 | 1 |
| MPNN | 91.3±4.1 | 80.8±2.4 | 69.1±3.0 | 59.5±3.0 | 87.9±5.4 | 81.5±1.0 | 75.7±1.3 | 77.0±1.4 |
| DMPNN | 91.9±3.0 | 75.9±0.7 | 63.7±0.2 | 57.0±0.7 | 90.6±0.6 | 85.2±0.6 | 78.6±1.4 | 77.1±0.5 |
| CMPNN | 92.7±1.7 | 80.1±1.6 | 70.8±1.3 | 61.6±0.3 | 89.8±0.8 | 86.7±0.2 | 79.0±2.0 | 78.2±2.2 |
| CoMPT | 96.1±0.4 | 84.5±0.7 | 72.2±0.8 | 66.1±0.9 | 97.3±2.5 | 94.1±3.6 | 82.6±1.6 | 86.4±1.2 |
| N-Gram | 91.2±0.3 | 76.9±2.7 | - | 63.2±0.5 | 87.5±2.7 | 79.1±1.3 | 76.9±0.7 | 78.7±0.4 |
| PretrainGNN | 70.8±1.5 | 78.7±0.4 | 65.7±0.6 | 62.7±0.8 | 72.6±1.5 | 84.5±0.7 | 81.3±2.1 | 79.9±0.7 |
| MGSSL | 70.5±1.1 | 76.4±0.4 | 64.1±0.7 | 61.8±0.8 | 80.7±2.1 | 79.7±0.8 | 78.7±1.5 | 79.5±1.1 |
| GEM | 88.8±0.4 | 78.1±0.4 | 68.6±0.2 | 63.2±1.5 | 90.3±0.7 | 87.9±1.1 | 75.3±1.5 | 81.3±0.3 |
| GROVER | 86.8±2.2 | 80.3±2.0 | 56.8±3.4 | 61.2±2.5 | 70.3±13.7 | 82.4±3.6 | 67.3±1.8 | 68.2±1.1 |
| GraphMVP | 72.4±1.6 | 75.9±0.5 | 63.1±0.4 | 63.9±1.2 | 79.1±2.8 | 81.2±0.9 | 77.7±0.6 | 77.0±1.2 |
| Uni-mol | 72.9±0.6 | 79.6±0.5 | 69.6±0.1 | 65.9±1.3 | 91.9±1.8 | 85.7±0.2 | 82.1±1.3 | 80.8±0.3 |
| DVMP | 77.8±0.3 | 79.1±0.4 | - | 69.8±0.6 | 95.6±0.7 | 89.4±0.8 | - | 81.4±0.4 |
| DVMP$_{MoleSG}$ | 80.9±2.1 | 84.4±1.2 | 73.3±0.9 | 66.9±1.2 | 98.4±2.0 | 93.5±2.8 | 80.9±2.1 | 87.6±1.8 |
| MolCLR | 73.3±1.0 | 74.1±5.3 | 65.9±2.1 | 61.2±3.6 | 89.8±2.7 | 82.8±0.7 | 78.9±2.3 | 77.4±0.6 |
| MolCLR$_{CoMPT}$ | 97.2±0.2 | 82.4±1.8 | 72.7±0.5 | 57.1±8.7 | 77.0±14.5 | 85.5±0.9 | 75.8±15.0 | 81.8±2.2 |
| KANO | 96.0±1.6 | 83.7±1.3 | 73.2±1.6 | 65.2±0.8 | 94.4±0.3 | 93.1±2.1 | 83.7±2.3 | 85.1±2.2 |
| MoleSG | **97.9±0.3** | **85.0±1.2** | **74.2±0.5** | **70.0±0.2** | **99.1±0.9** | **95.1±2.1** | **85.1±0.8** | **87.7±1.9** |

Table 2: Performance of different models on six regression benchmarks in physical chemistry and quantum mechanics. The mean and standard deviation of root mean square error (RMSE) (for ESOL, FreeSolv, and Lipophilicity) or mean absolute error (MAE) (for QM7, QM8, and QM9) from three independent runs are reported. (Lower values indicate better performance.)

| Category | Physical chemistry | | | Quantum mechanics | | |
|---|---|---|---|---|---|---|
| Dataset | ESOL | FreeSolv | Lipophilicity | QM7 | QM8 | QM9 |
| Molecules | 1128 | 642 | 4200 | 6830 | 21786 | 133885 |
| Tasks | 1 | 1 | 1 | 1 | 12 | 3 |
| MPNN | 1.167±0.043 | 1.621±0.952 | 0.672±0.051 | 111.4±0.9 | 0.0148±0.001 | 0.00522±0.00003 |
| DMPNN | 1.050±0.008 | 1.673±0.082 | 0.683±0.016 | 103.5±8.6 | 0.0156±0.001 | 0.00514±0.00001 |
| CMPNN | 0.798±0.112 | 1.570±0.442 | 0.614±0.029 | 75.1±3.1 | 0.0153±0.002 | 0.00405±0.00002 |
| CoMPT | 0.643±0.051 | 0.970±0.207 | 0.572±0.058 | 32.7±7.4 | 0.0120±0.001 | 0.00353±0.00067 |
| N-Gram | 1.100±0.030 | 2.510±0.191 | 0.880±0.121 | 125.6±1.5 | 0.0320±0.003 | 0.00964±0.00031 |
| PretrainGNN | 1.100±0.006 | 2.764±0.002 | 0.739±0.003 | 113.2±0.6 | 0.0215±0.001 | 0.00922±0.00004 |
| GEM | 0.813±0.028 | 1.748±0.114 | 0.674±0.022 | 60.0±2.7 | 0.0163±0.001 | 0.00562±0.00007 |
| GROVER | 1.423±0.288 | 2.947±0.615 | 0.823±0.010 | 91.3±1.9 | 0.0182±0.001 | 0.00719±0.00208 |
| Uni-mol | 0.788±0.029 | 1.480±0.048 | 0.603±0.010 | 41.8±0.2 | 0.0156±0.000 | - |
| DVMP | 0.817±0.024 | 1.952±0.061 | 0.653±0.002 | 74.4±1.2 | 0.0171±0.004 | - |
| DVMP$_{MoleSG}$ | 0.669±0.114 | 0.942±0.110 | 0.594±0.018 | 30.2±3.0 | 0.0123±0.001 | 0.00323±0.00006 |
| MolCLR | 1.113±0.023 | 2.301±0.247 | 0.789±0.009 | 90.9±1.7 | 0.0185±0.013 | 0.00480±0.00003 |
| MolCLR$_{CoMPT}$ | 0.849±0.062 | 1.135±0.163 | 0.657±0.012 | 32.7±2.8 | 0.0141±0.001 | 0.00350±0.00000 |
| KANO | 0.670±0.019 | 1.142±0.258 | 0.566±0.007 | 56.4±2.8 | 0.0123±0.000 | 0.00320±0.00001 |
| MoleSG | **0.599±0.067** | **0.932±0.131** | **0.545±0.014** | **29.6±2.9** | **0.0117±0.001** | **0.00313±0.00006** |

Table 3: Comparison of our approach with two single-modality pre-training approaches on classification tasks. The mean and standard deviation of ROC-AUC (%) over three independent runs are reported. (Higher values indicate better performance.)

| | BBBP | Tox21 | ToxCast | SIDER | Clintox | BACE | MUV | HIV |
|---|---|---|---|---|---|---|---|---|
| SMILES scratch | 63.6±4.3 | 75.5±0.5 | 64.2±2.5 | 54.0±2.4 | 88.1±6.3 | 79.2±6.6 | 63.6±4.3 | 72.7±3.5 |
| SMILES pre-train | 61.5±4.9 | 77.6±2.5 | 66.8±0.9 | 55.0±3.1 | 93.3±2.8 | 83.8±0.9 | 61.5±4.9 | 75.1±2.5 |
| Ours SMILES | **65.3±3.1** | **77.9±2.5** | **67.0±0.9** | **59.6±3.8** | **94.3±2.0** | **85.3±1.1** | **65.3±3.1** | **77.3±0.7** |
| Graph scratch | 96.1±0.4 | 84.5±0.7 | 72.2±0.8 | 66.1±0.9 | 97.3±2.5 | 94.1±3.6 | 82.6±1.6 | 86.4±1.2 |
| Graph pre-train | 96.8±1.8 | 84.2±0.1 | 72.6±1.0 | 66.7±2.2 | 98.0±0.9 | 94.9±2.3 | 82.2±1.4 | 85.9±2.5 |
| Ours graph | **97.9±0.3** | **85.0±1.2** | **74.2±0.5** | **70.0±0.2** | **99.1±0.9** | **95.1±2.1** | **85.1±0.8** | **87.7±1.9** |

Table 4: Comparison of our approach with two single-modality pre-training approaches on regression tasks. The mean and standard deviation of RMSE or MAE over three independent runs are reported. (Lower values indicate better performance.)

| | ESOL | Freesolv | Lipophilicity | QM7 | QM8 | QM9 |
|---|---|---|---|---|---|---|
| SMILES scratch | 0.946±0.226 | 2.581±0.286 | 1.028±0.030 | 160.2±6.8 | 0.0146±0.001 | 0.01017±0.00045 |
| SMILES pre-train | 1.030±0.336 | 1.942±0.450 | 1.034±0.015 | 159.3±5.7 | 0.0141±0.001 | 0.01080±0.00010 |
| Ours SMILES | **0.873±0.172** | **1.889±0.590** | **0.964±0.036** | **155.7±3.9** | **0.0139±0.001** | **0.00973±0.00059** |
| Graph scratch | 0.643±0.051 | 0.970±0.207 | 0.572±0.058 | 32.7±7.4 | 0.0120±0.001 | 0.00353±0.00067 |
| Graph pre-train | 0.635±0.104 | 0.939±0.225 | 0.585±0.031 | 32.3±1.6 | 0.0118±0.001 | 0.00323±0.00012 |
| Ours graph | **0.599±0.067** | **0.932±0.131** | **0.545±0.014** | **29.6±2.9** | **0.0117±0.001** | **0.00313±0.00006** |

of the two modalities in MoleSG contributes to outstanding results, surpassing methods injecting additional 3D information.

Table 2 shows the test results in regression tasks. We can observe that MoleSG achieves the best scores among both supervised and self-supervised pre-training models, with a relative improvement of 14.4% over KANO across all six regression tasks. MoleSG greatly benefits tasks with limited label information, achieving a 18.4% improvement over KANO on the small dataset FreeSolv, which contains only 642 labeled molecules.

Moreover, it is worth noting that our method still outperforms MolCLR$_{\text{CoMPT}}$, which is a version of the typical contrastive learning method MolCLR with the same encoder as ours, verifying the superiority of our method. We also compare with another contrastive learning competitor DVMP$_{\text{MoleSG}}$, which utilizes the same encoders as ours. In addition, both MolCLR$_{\text{CoMPT}}$ and DVMP$_{\text{MoleSG}}$ outperform their original counterpart MolCLR and DVMP in most tasks, demonstrating the effectiveness of the corresponding strategies proposed in this paper.

### 4.3 ABLATION EXPERIMENTS

#### 4.3.1 SINGLE-MODALITY VS. MULTI-MODALITY

To further reveal the superiority of our method, we compare our multi-modality pre-training with single-modality pre-training. The results are shown in Table 3 and Table 4. Our method successfully achieves the best performance on all downstream tasks. Moreover, it is worth noting that single modality pre-training may cause performance degradation. However, by fully leveraging the complementary information among different modalities, our method can improve performance on all downstream tasks, showing more potential for practical applications. We present visualization results of our method's feature extraction capability in Appendix D.

#### 4.3.2 OVERLAP VS. NON-OVERLAP

To validate whether our non-overlapping masking strategy benefits pre-training, we conduct experiments on different overlap ratios on all downstream tasks. We define overlap ratio as a metric measuring the proportion of jointly masked atoms in both modality inputs. We conduct experiments at overlap ratios at 0%, 25%, 50%, 75%, and 100% across all benchmarks, where our non-overlapping

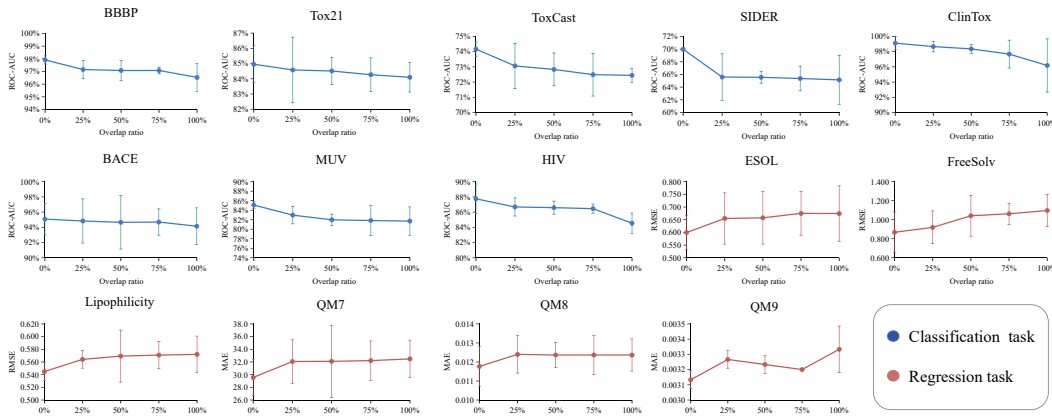

Figure 4: The impact of different overlap ratios on downstream task performance. The results are reported as mean and standard deviation values on three independent runs.

masking strategy is equivalent to setting the overlap ratio to 0. The experimental results shown in Figure 4 indicate that the performance on downstream tasks is the best when the overlap ratio is 0.

### 4.3.3 WITH VS. WITHOUT BACKBONE

As analyzed above, fine-tuning both the encoder and backbone may cause suboptimal performance due to the inconsistent distributions. Therefore, we conduct an experiment to validate it. Specifically, section 4.3.1 has shown that the graph encoder has better performance than the SMILES encoder. Therefore, we only consider two combinations in this section. The former is fine-tuning a single graph encoder, and the other is fine-tuning both the graph encoder and the backbone. We perform experiments on all benchmarks, and the results are shown in Table 5 and Table 6. The results show that using only the graph encoder achieves higher performance in all tasks.

Table 5: Comparison of results on classification tasks with and without the backbone network. The mean and standard deviation of ROC-AUC (%) from three independent runs are reported.

|  | BBBP | Tox21 | ToxCast | SIDER | ClinTox | BACE | MUV | HIV |
|---|---|---|---|---|---|---|---|---|
| Graph encoder+backbone | 97.23±0.6 | 84.8±1.8 | 73.6±0.9 | 65.6±0.4 | 98.8±0.6 | 89.7±5.2 | 81.9±1.9 | 85.8±1.4 |
| Graph encoder | **97.9±0.3** | **85.0±1.2** | **74.2±0.5** | **70.0±0.2** | **99.1±0.9** | **95.1±2.1** | **85.1±0.8** | **87.7±1.9** |

Table 6: Comparison of results on regression tasks with and without the backbone network. The mean and standard deviation of RMSE (or MAE) from three independent runs are reported.

|  | ESOL | FreeSolv | Lipophilicity | QM7 | QM8 | QM9 |
|---|---|---|---|---|---|---|
| Graph encoder+backbone | 0.661±0.011 | 0.988±0.250 | 0.560±0.017 | 31.9±3.8 | 0.0119±0.001 | 0.00353±0.00015 |
| Graph encoder | **0.599±0.067** | **0.932±0.131** | **0.545±0.014** | **29.6±2.9** | **0.0117±0.001** | **0.00313±0.00006** |

## 5 CONCLUSION

In this study, we address the challenges of learning fine-grained information from two complementary modalities: SMILES and graph. To better capture rich molecular features from the interaction between these two modalities, we design a simple and efficient multi-modality pre-training framework called MoleSG, which utilizes a unified feature processing network to fuse both modalities. In addition, we propose a non-overlapping masking strategy to facilitate information exchange between the two modalities. Extensive experiments on 14 downstream tasks show that our method achieves new SOTA performance. Our non-overlapping masking strategy has the potential to be used in other masked reconstruction-based multi-modality pre-training studies.

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
