# A    DOWNSTREAM DATA SUPPLEMENTS

The 14 downstream task datasets are sourced from MoleculeNet (Wu et al., 2018) and can be categorized into physiology, biophysics, physical chemistry, and quantum mechanics. The details are as follows:

**Physiology:** BBBP (Martins et al., 2012) comprises information concerning whether a compound exhibits the capability to traverse the blood-brain barrier. Tox21 (Hartung, 2009) is a publicly accessible database designed to assess the toxicity profiles of various compounds, notably in the 2014 Tox21 Data Challenge. ToxCast (Richard et al., 2016) houses an extensive array of toxicity labels for thousands of compounds, derived from high-throughput screening tests on a vast chemical library. SIDER (Kuhn et al., 2016) archives information on commercially available medications, complete with details on their associated adverse drug reactions. ClinTox (Gayvert et al., 2016) conducts a comparative analysis between drugs that have received FDA approval and those that have been eliminated during clinical trials due to safety concerns.

**Biophysics:** BACE (Subramanian et al., 2016) serves as a repository for compounds identified in recent years as potential inhibitors of human $\beta$-secretase 1 (BACE-1). MUV (Rohrer & Baumann, 2009) represents a refined subset of the PubChem BioAssay, specifically tailored for the validation of virtual screening techniques through advanced nearest neighbor analysis. HIV (Riesen & Bunke, 2008) provides experimental data on the inhibitory capabilities of over 40,000 molecules against HIV replication.

**Physical Chemistry:** ESOL (Delaney, 2004) is a compact dataset documenting compound solubility. FreeSolv (Mobley & Guthrie, 2014) is derived from the Free Solvation Database, containing hydration-free energy data for small molecules. Lipophilicity (Gaulton et al., 2012) is sourced from the ChEMBL database and contains experimental octanol-water partition coefficient results.

**Quantum Mechanics:** QM7 (Blum & Reymond, 2009) provides molecular spatial structure information and stable, synthetically obtainable electronic properties such as HOMO, LUMO, and atomization energy, determined using ab-initio density function theory (DFT). QM8 (Ramakrishnan et al., 2015) employs various quantum mechanics methods to compute electronic spectra and excited state energies for small molecules. QM9 (Ruddigkeit et al., 2012) offers extensive data on geometry, energy, electronic, and thermodynamic properties of small molecules calculated via DFT.

Table 7 provides detailed information on these 14 datasets, including task types, evaluation metrics, molecular categories, data size, and split types. As shown in Table 7, we employ scaffold splitting for all benchmarks except QM9. Scaffold splitting in chemical datasets is valuable for enhancing the generalization performance of machine learning models by ensuring diverse molecular structures in both training and test sets. This strategy promotes more realistic performance assessments and mitigates the risk of overfitting, ultimately facilitating reliable model selection and optimization in chemical informatics and drug discovery applications. For QM9, we use random splitting based on previous research (Wang et al., 2022; Fang et al., 2023). Our evaluation metric for classification tasks is ROC-AUC, while for regression tasks, we employ RMSE and MAE as our evaluation metrics.

# B    COMPETITORS

To verify MoleSG's effectiveness, we conduct a thorough performance evaluation, comparing it with supervised and self-supervised learning competitors.

**Competitors experimental setup:** In this paper, we compare MoleSG with 14 baseline methods, including MPNN (Gilmer et al., 2017), DMPNN (Yang et al., 2019), CMPNN (Song et al., 2020), CoMPT (Chen et al., 2021), N-gram (Liu et al., 2019a), PretrainGNN (Hu et al., 2019), MGSSL (Zhang et al., 2021), GROVER (Rong et al., 2020), GraphMVP (Liu et al., 2021), MolCLR (Wang et al., 2022), GEM (Fang et al., 2022), DVMP (Zhu et al., 2021), KANO (Fang et al., 2023), and Uni-mol (Zhou et al., 2023). The results of MPNN, DMPNN, CMPNN, CoMPT, N-gram, PretrainGNN, MGSSL, GROVER, GraphMVP, MolCLR, GEM, and KANO are taken from the paper of KANO (Fang et al., 2023), while the results of DVMP and Uni-mol are obtained from there original articles. To ensure a fair comparison, we adhere to the experimental setup established in prior research. This setup involves conducting experiments with three independent train/val/test data splits. The objective is to assess the models' robustness and reduce the potential influence of variations in

Table 7: The detailed information of all the benchmarks for molecular property predictions used in this work. The benchmarks contain eight graph classification datasets and six graph regression datasets.

| Dataset | Task Type | Metric | Category | Tasks | Compounds | Split |
|---|---|---|---|---|---|---|
| BBBP | | | | 1 | 2039 | |
| Tox21 | | | | 12 | 7831 | |
| ToxCast | | | Physiology | 617 | 8575 | |
| SIDER | Classification | ROC-AUC | | 27 | 1427 | |
| ClinTox | | | | 2 | 1478 | |
| BACE | | | | 1 | 1513 | |
| MUV | | | Biophysics | 17 | 93087 | scaffold split |
| HIV | | | | 1 | 41127 | |
| ESOL | | | | 1 | 1128 | |
| FreeSolv | | RMSE | Physical chemistry | 1 | 642 | |
| Lipophilicity | Regression | | | 1 | 4200 | |
| QM7 | | | | 1 | 6830 | |
| QM8 | | MAE | Quantum mechanics | 12 | 21786 | |
| QM9 | | | | 3 | 133885 | random split |

model performance stemming from different data splits. The experimental settings for downstream tasks are the same as those used in CoMPT (Chen et al., 2021).

## C    MASK RATIO SETUP

To determine the mask ratio for graph and SMILES modalities, we use a controlled variable approach. We adjust the graph mask ratio while keeping the SMILES mask ratio constant, and vice versa. We conduct experiments across all benchmarks, and the experimental results for the SMILES mask ratio and graph mask ratio are shown in Figure 5 and Figure 6, respectively. We observe that a graph mask ratio of 25% and a SMILES mask ratio of 15% are suitable for our purposes. (In classification tasks, a higher ROC-AUC(%) value indicates better performance, while in regression tasks, lower RMSE and MAE values are desirable.)

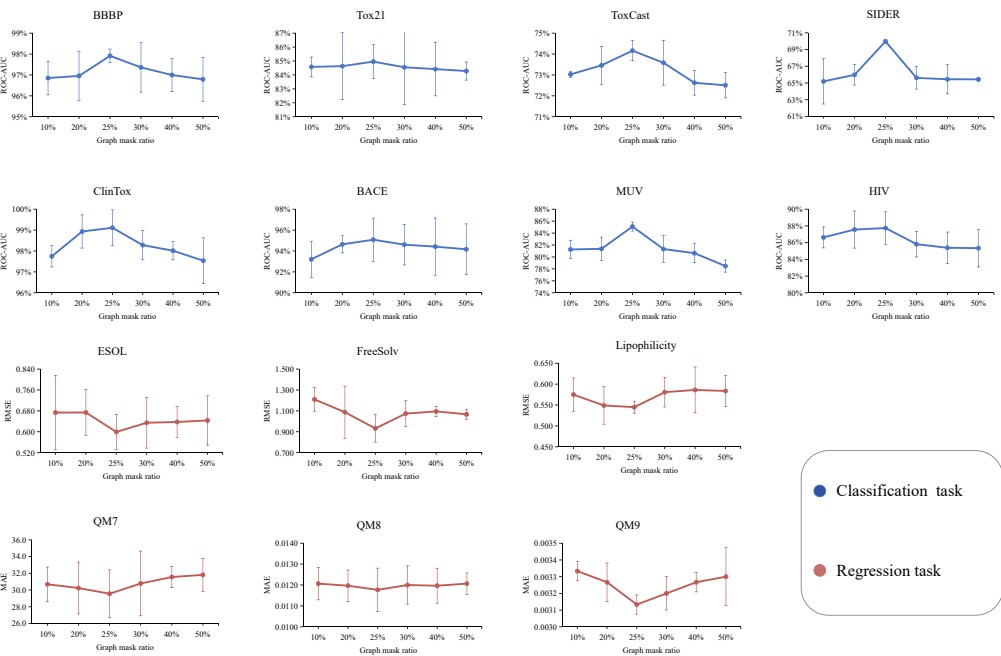

Figure 5: The impact of different mask ratios on downstream task performance on graph. The results are reported as mean and standard deviation values on three independent runs.

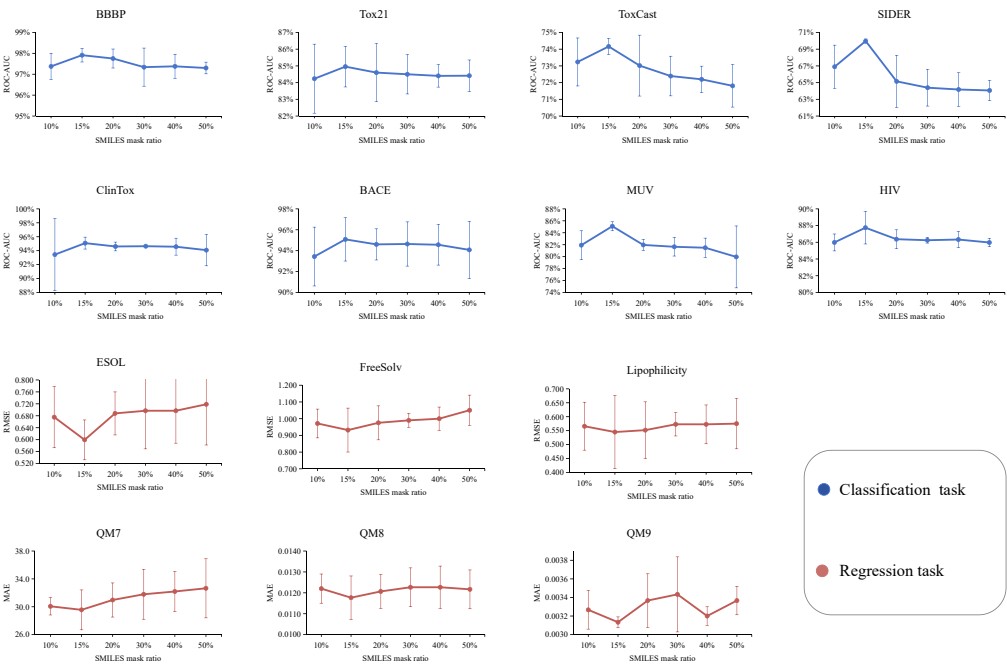

Figure 6: The impact of different mask ratios on downstream task performance on SMILES. The results are reported as mean and standard deviation values on three independent runs.

# D    VISUALIZATION

Figure 7 illustrates the strong feature discriminative ability of MoleSG in the classification tasks BBBP and BACE. We compare models without pre-training, single-modality pre-training (i.e., graph pre-training), and contrastive pre-training (DVMP$_{MoleSG}$). We can observe the superior feature discrimination of our approach compared to single-modality pre-training and contrastive pre-training.

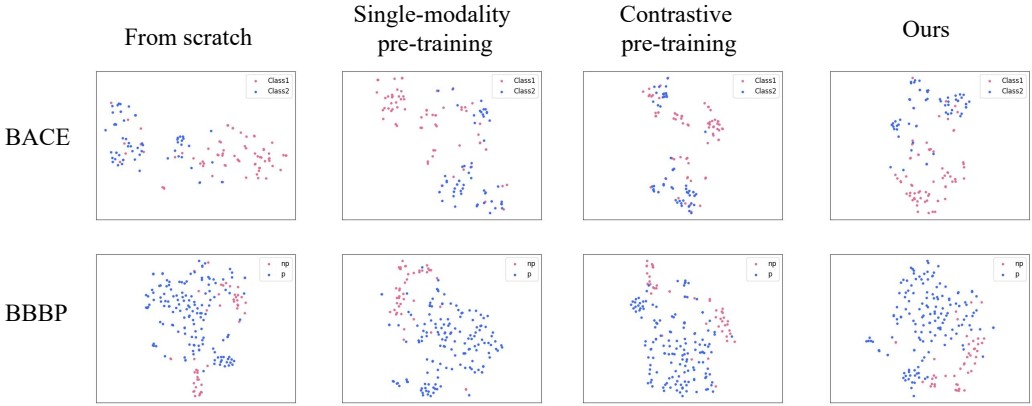

Figure 7:   T-SNE visualization of feature separation of four methods on BACE and BBBP benchmark datasets.

# E    TOKEN VOCABULARY

A Simplified Molecular Input Line Entry System (SMILES) is a linear notation used to represent molecules in a simple and compact way, categorizing their components into three types: atoms, bonds, and other representations encoding ring closures in the graph. An example of a molecule represented in SMILES notation is shown in Figure 1, where the SMILES representation for a molecule with the structure c1cc(F)ccc1Cl is provided alongside its 3D molecular structure. In summary, in the SMILES notation, characters like C, Cl, and F typically signify atoms, while symbols such as '-', '=', and '' represent chemical bonds, and numeric values indicate adjacent atoms in ring-closing segments of the molecule. However, it's important to acknowledge that the SMILES system doesn't always offer a perfect one-to-one correspondence between SMILES sequences and molecular structures. For instance, a molecule may have multiple equivalent SMILES representations, such as C(F)C, FCC, and CCF. To address this challenge and ensure a unique mapping between SMILES and molecules, various standardization algorithms have been developed, guaranteeing the representation's consistency. For this study, all employed SMILES representations adhere to standardized conventions.

Below, we will list all the elements found in the SMILES data, along with their corresponding token IDs. Having knowledge of all the elements indicated by these IDs will facilitate the extraction of atomic representations for our non-overlapping masking strategy.

Token vocabulary:'[13cH]': 279, '[N-3]': 294, '[NH3+2]': 510, '[nH]': 40, '[O+]': 73, '[Br]': 218, '[Th+4]': 534, '[Mn-2]': 682, '[S-2]': 201, '[123I-]': 368, '[NH-]': 161, '[Mg]': 77, '[Nd+]': 412, '[13CH3]': 260, '[Sb]': 200, '[BrH+]': 479, '[35S]': 497, '[Ca-2]': 624, '[YH]': 372, '[Si@]': 268, '[Tl+]': 304, '[Mn+2]': 190, '[2NH]': 588, '[Li-]': 538, 's': 42, '[Se-]': 321, '[3H]': 257, '[SH5]': 504, '[18F-]': 340, '[HgH]': 649, '[BH2]': 442, '[unused10]': 10, '[Ru]': 117, '[Os+4]': 306, '[Co+]': 500, '[11c]': 548, '[Mo-]': 641, '[Si+4]': 245, '[PH2]': 164, '[N+]': 41, ')': 18, '[Cu+]': 131, '[Ra]': 658, '5': 43, '[c+]': 314, 'b': 410, '[N@@H+]': 486, '[Ni-3]': 634, '[NH4]': 594, '[Fm]': 415, '[SH]': 139, '[AlH2]': 377, '[SnH4]': 342, '[c]': 466, '[BaH]': 531, '[unused3]': 3, '[11CH3]': 284, '[13C]': 228, '[Nd+3]': 277, '[18OH2]': 557, '[C@@]': 56, '[Cs+]': 68, '[Si+2]': 567, '[201Tl]': 595, '': 38, '[Si]': 47, '[Mo+2]': 470, '[13N]': 586, '[C@@H]': 35, '[AlH4-]': 103, '[Er]': 424, '[pH+]': 666, '[Rh+2]': 199, '[PH3]': 329, '[Zr-2]': 501, '[OH-]': 59, '[Zn-4]': 621, '(': 17, 'Br': 37, '[Hg+2]': 186, '[PH4]': 452, '[V+5]': 251, '[In]': 192, '[SH2]': 324, '[Ir+2]':

438, '[15NH2]': 336, '[9CH]': 563, '[As+]': 443, '[Ba]': 207, '-': 31, '[C@H]': 33, '[Ga+]': 429, '[123I]': 313, '[Mo]': 154, '[15OH2]': 587, '[CH]': 83, '[Co+2]': 182, '%12': 202, '[Hg+]': 283, '[PbH2+2]': 647, '[Al-]': 225, '[V]': 223, '[Tl+2]': 518, '[Fe]': 93, '[SiH-]': 673, '[211At]': 509, '[AsH]': 474, '[CH2-]': 157, '[Sn+2]': 168, '[se+]': 439, '[Zn]': 87, '[Ru-4]': 620, '[o+]': 215, '[SeH+]': 434, '%19': 388, '[Cu-]': 224, '[IH2+]': 330, '[Ge]': 209, '[Mo+4]': 382, '[C+4]': 393, '[153Sm]': 561, '[Ti+2]': 269, '[Cd+2]': 309, '[Be+2]': 392, 'p': 206, '[B+2]': 506, '[AsH4]': 689, '[Hf]': 285, '[Li+]': 69, '[Cu-4]': 602, '[14CH2]': 555, '[Sn-]': 669, '[SbH6+3]': 644, '[Pd-2]': 273, '[Sm+3]': 325, '[PH-]': 459, '[GeH2]': 467, '[Dy]': 473, '[SbH]': 487, '[13CH]': 423, '[VH]': 505, '[PH]': 115, '[SnH]': 150, '[Na-]': 454, '[13c]': 227, '[Pa]': 651, '[Ni-4]': 639, ' ': 30, '[Cs]': 70, '[Ni+2]': 180, '[Rb+]': 315, '[No]': 665, '[Ni-2]': 631, '[Co-4]': 611, '[Ac-]': 558, '[Ag]': 123, '[SH-]': 173, 'F': 27, '[W]': 149, '[Cd]': 289, '[Yb]': 335, '[TeH]': 343, '[Mn+3]': 320, '[Nd]': 359, '%17': 370, 'o': 44, '[2H]': 64, '[Zr+]': 413, '[Hf+3]': 420, '[In+]': 477, '%13': 217, '[60Co]': 488, '[TlH]': 650, '[SEP]': 13, '[NH2+]': 116, '[Ga-3]': 625, '[Ca-4]': 610, '[Dy+3]': 418, '[Y]': 287, '[unused1]': 1, '[Bi+5]': 571, '%16': 355, '[P-3]': 577, '[Te]': 226, '[IH4]': 687, '[Ac]': 130, '[Cr+3]': 211, '[Sr+2]': 246, '[Ag+]': 118, '[14c]': 369, '[U+2]': 396, '[Er+3]': 367, '[Cr-]': 514, '8': 98, '[Sn+4]': 282, '[SH+]': 236, '%21': 398, '[15N]': 404, '[Cn]': 590, '[UNK]': 11, '[V+]': 608, '[W+2]': 601, '[NH+3]': 536, '[Fe-]': 683, '[Ir-3]': 656, '[Pt]': 109, '[Si-]': 250, '[CaH2]': 346, '[Ti+5]': 496, '[Au]': 212, '[B+]': 305, '[Yb+3]': 187, '[Ca+2]': 122, '[Ag+3]': 449, '%18': 379, '[Sm]': 255, '[TeH3]': 447, '[Mn+6]': 568, '[B-2]': 633, '[SiH3]': 151, '*': 256, '[125I]': 363, '[CH3+]': 317, '[SnH2]': 371, 'I': 48, '[unused4]': 4, '[S@+]': 220, '[SiH4+]': 556, '[AlH3-3]': 640, '[TlH2]': 664, '[67Ga+3]': 584, '[cH-]': 135, '[7NaH]': 539, '[OH+]': 261, '[Pb+2]': 174, '[Mo-3]': 604, '[CH-]': 188, '[Ag+2]': 318, '[Ca]': 128, '[Cu-2]': 276, '[ZrH2]': 569, '[Ba+]': 475, '[SnH2+2]': 646, '[n+]': 79, '[S@]': 108, '[SeH]': 249, '[Ni+]': 446, '[AsH3]': 490, '[Fe+3]': 152, '[NH2]': 258, '%10': 156, '[64Cu]': 519, '[1H]': 522, '[NH+]': 121, '[Rh+]': 292, '[Si@@H]': 344, '[Tl]': 347, '[PH+]': 143, '[C-4]': 358, '[Cm]': 430, '[Ir-]': 391, '[Cf]': 444, '[Cl-]': 57, '[Cr+]': 471, '[U-5]': 606, '[Pd-3]': 600, '[124I-]': 578, '[FH+]': 618, '[Pr]': 286, '[GaH3]': 605, '[BH]': 384, '[Cr]': 105, '[S+4]': 311, '[Mg+]': 147, '[249Cf]': 642, '[Se]': 126, '[U+6]': 433, '[Ni+3]': 516, '[Rh-3]': 216, '[IH-]': 432, 'c': 15, '[(1R)-1-methylpropyl]': 597, '[I-]': 84, '[Cl+3]': 134, '[Sb+3]': 297, '[PH5]': 354, '[68Ga]': 458, '[Hg-2]': 627, '[Zr]': 195, '[Zn+2]': 114, '[cH+]': 319, '[15NH]': 395, '[Mo+6]': 528, '[Zr+2]': 189, '[Pd-]': 308, '[Nb+3]': 581, '[32P]': 435, '[TlH2+]': 643, '[AlH3-]': 630, '[Zn-3]': 622, '[S@@+]': 234, '[RuH2]': 394, '[Ti+3]': 198, '[19F]': 378, '[18O]': 451, '[Pt-]': 437, '[AsH+]': 529, '[Fe+4]': 532, '[Ar]': 222, '[unused9]': 9, '[P+]': 91, '%11': 170, '[13CH2]': 265, '[se]': 142, '[Si@@]': 267, '[Zr+3]': 291, '[Ta+5]': 348, '[11C]': 422, '[H]': 63, '[KH]': 193, '[Rb]': 386, '[Ce+2]': 461, '[Sn+3]': 365, '[IH+]': 326, '[PtH+2]': 541, '[Pt-4]': 617, '[Mn]': 100, '[Cu-5]': 619, '': 592, '[131I]': 503, '[Nb+4]': 580, '[Br+2]': 238, '[Se+]': 575, '[Co]': 162, '[AsH4+]': 489, '[Co+3]': 310, '[Ru+2]': 300, '[SiH4]': 231, '[S-]': 112, '[Ho+3]': 457, '[Re+]': 612, '[Si-2]': 645, '[(2S)-butan-2-yl]': 598, '[PAD]': 0, '[W+4]': 421, '[Cu+3]': 469, '[asH]': 674, '[Ru-]': 527, '[WH]': 690, '[Al+2]': 402, '[Co-3]': 660, '[N@@]': 242, '[N-]': 61, '[Tl+3]': 352, '[K]': 52, '[Eu+3]': 323, 'Cl': 28, '[Al+3]': 110, '[Be]': 385, '[IH]': 361, '[CuH2-]': 579, '[Sn]': 80, '[n-]': 159, '[229Th]': 533, '[HeH]': 551, '[Zr+4]': 176, '[Sb+2]': 455, '[IH2]': 686, '[Ir-4]': 637, '[V+2]': 316, '[H+]': 140, '[Ag-]': 259, '[Gd+3]': 264, '[CH2+]': 155, '[SH2+]': 331, '[La+3]': 240, '[Nb]': 366, '[Na]': 49, '[pH]': 387, '[PH2+]': 414, '[nH+]': 102, '[In+2]': 484, '[unused5]': 5, '[Ir]': 178, '[At]': 576, '[13C@@H]': 464, '%14': 298, '[Ir+4]': 583, '[MgH2]': 629, '[Re]': 253, '[Zn+]': 136, '[31P]': 495, '[Rh-]': 513, '[C-2]': 570, '[Al]': 90, '[N@H+]': 416, '[Gd]': 339, '[b-]': 636, '[NaH]': 71, '[Rh+3]': 235, '[Ta]': 303, '[AlH4]': 95, '[SbH2]': 448, '[CuH2]': 494, '[Sc+2]': 552, '[PdH2]': 307, '[15n]': 401, '[OH3+]': 328, '[p+]': 662, '[I+]': 163, '[Nb+5]': 312, '[Br-]': 85, 'P': 45, '[O-2]': 104, 'N': 23, '[I+3]': 120, '[SiH2]': 127, '[F]': 427, '[BH2-]': 502, '[Ni-]': 545, '[IH+3]': 677, '[Cu+2]': 99, '[PH4+]': 230, '[Al+]': 247, '¡s¿': 591, '[Bi+]': 540, '[ArH]': 499, '[15CH]': 574, '[Xe]': 390, '[Os-3]': 653, '[BiH2]': 685, '[BrH2+]': 628, '[10B]': 468, '[Pb+4]': 210, '[14cH]': 480, '[NiH2]': 566, '%15': 327, 'S': 34, 'B': 54, '[P@]': 181, '[ghi]': 596, '[W+]': 481, '2': 21, '[Tl-3]': 613, '[Ti]': 113, '[Ni]': 96, '[Cd+]': 521, '[Hf+2]': 295, '[Tb+3]': 403, '[9CH3]': 564, '[P@H]': 537, '[B+3]': 275, '[AlH2-]': 196, '[NH]': 153, '[MASK]': 14, '[Ce+3]': 194, '[Lu]': 337, '[V-]': 615, '[Zn-2]': 609, '[S+]': 119, '\\': 60, '[Randic connectivity]': 593, '[Eu]': 364, '[Pd-4]': 638, '[Am]': 670, '[C+]': 146, '[BH3-]': 92, '[Tb]': 380, '[Cu-3]': 623, '[Sn+]': 270, '[Ga-]': 676, '[SH3+]': 341, '[Tc]': 520, '[Ir-2]': 678, '[Fe+]': 655, '[N@@+]': 356, '[Y+3]': 252, '[P]': 53, '[PbH]': 659, '[Ba+2]': 169, '[Na-2]': 493, '[Si@H]': 204, '[BH4-]': 74, '%20': 389, '[NH2-]': 208, '[Y-]': 565, '[I]': 144, '[SnH3]': 274, '[GeH]': 349, '[Te+]': 667, '[P+3]': 185, '[As+3]': 535, '[O]': 137, '[N@]': 244, '[IrH]': 511, '[Pb]': 248, '[Xe+]': 546, '[SeH-]':

436, '[unused2]': 2, '[Os-2]': 175, '[Au+3]': 357, '[14C@H]': 508, '[Fe+6]': 465, 'C': 16, '.': 24, '6': 58, '[Pt+4]': 165, '[Pd]': 50, '[H-]': 72, '[Pt+2]': 145, '[Pt-2]': 241, '[V+3]': 405, '[W+6]': 406, '[13C@H]': 482, '[Pd+4]': 472, '[24NaH]': 572, '[AlH]': 101, '[NH4+]': 65, '[Pd+2]': 82, '[18OH]': 491, '%23': 419, '[UH]': 554, '[SeH2]': 476, '7': 76, '[LiH]': 213, '[PbH2]': 654, '[15nH]': 523, '[Cl+2]': 271, '[BaH2]': 530, '[Rh+4]': 549, '[Mo+]': 663, '[CH2]': 158, '[La]': 239, '[Li]': 62, '[Al-2]': 671, '[PH3+]': 243, '[99Tc]': 463, '[TaH3]': 517, '[GeH2+]': 614, '[C@]': 55, 'O': 19, '[Cr+2]': 299, '[Rh]': 133, '[AsH2]': 483, '[p-]': 542, '[67Ga]': 585, '[IH3]': 684, '[Ru+3]': 262, '[Si+]': 383, '[Au+]': 400, '[Rh-4]': 626, '3': 26, '[GeH3]': 431, '[AlH-]': 272, 'n': 25, '[N@+]': 362, '[Cr+6]': 290, '[Bi]': 179, '[as]': 657, '[Th]': 296, '[C-]': 86, '[N+2]': 485, '[Re+5]': 543, '[Sc+3]': 214, '[Mg+2]': 106, '[Ho]': 589, '[Sb-]': 219, '[S]': 132, '[te]': 232, '[Hf+4]': 301, '[K+]': 51, '[Ti+]': 360, '[Co-2]': 616, '[SiH]': 97, '[Cu]': 75, '[Ir+]': 375, '[Hg]': 148, '[siH]': 428, '[Cr+4]': 408, '9': 124, '[In-]': 675, '[14C]': 302, '[Sb+]': 515, '[Cl]': 229, '[Hg-]': 668, '[S+2]': 288, '[Sm+2]': 353, '[Mn+]': 553, '[AcH]': 550, '[ClH+]': 409, '[c-]': 66, '[Os]': 125, '[sH+]': 338, '[B-]': 88, '[Fe+2]': 78, '[N]': 177, '[CH+]': 191, '[Ti-2]': 632, '[OH2+]': 197, '4': 32, '[Sc]': 334, '[FeH]': 680, '[RuH]': 417, '/': 39, '[As]': 184, '[unused8]': 8, '[Ce]': 166, '[Na+]': 46, '[Sr]': 351, '[I+2]': 221, '[Sb+5]': 373, '[N+3]': 376, '[Ti+4]': 129, '[Ce+4]': 160, '[BH-]': 81, '[Br+]': 350, '[14CH]': 381, '[Ti+6]': 411, '[BiH3]': 526, '[PH2-]': 688, '[P-]': 67, '[As-]': 322, '[14CH3]': 407, '[U]': 440, '[Bi+2]': 460, '[Ru-2]': 492, '[Yb+2]': 507, '1': 20, '[Ca+]': 547, '[FH+2]': 525, '[XeH]': 652, '[IH2+3]': 681, '[C]': 94, '[In+3]': 266, '[CH3]': 293, '[IrH2]': 512, '[Pt-3]': 603, '[AlH2+]': 524, '[ClH2+]': 445, '[V+4]': 456, '[Ru+]': 254, '[CLS]': 12, '[MgH]': 582, '[Au-3]': 635, '[Fe-3]': 171, '[s+]': 183, '[Au-]': 397, '[Ta-]': 679, '[P@@]': 203, '>>': 29, '[118Sn]': 560, '[Mn+4]': 205, '[CH3-]': 233, '[Cl+]': 138, '[124I]': 425, '[P@@H]': 498, '[Pd+]': 141, '[LaH]': 426, '%24': 450, '[Pr+3]': 374, '[Pm]': 672, '[F+]': 441, '[TeH2]': 478, '[18F]': 172, '[Pb+3]': 278, '[P+2]': 562, '[Ga]': 281, '[unused6]': 6, '[2-benzhydryloxyethyl]': 599, '[Ir+3]': 332, '[unused7]': 7, '=': 22, '[O-]': 36, '[Ga+3]': 345, '[B]': 167, '[Os+2]': 453, '[Fe-2]': 661, '[AlH3]': 333, '%22': 399, '[NiH]': 544, '[S@@]': 111, '[OH]': 263, '[Re+4]': 559, '[Bi+3]': 237, '[F-]': 89, '[Fe-4]': 280, '[Pd+3]': 462, '[Fr]': 573, '[NH3+]': 107, '[Po]': 648, '[Tl-]': 607.