# OpenReview forum: "MoleSG: A Multi-Modality Molecular Pre-training Framework by Joint Non-overlapping Masked Reconstruction of SMILES and Graph"
_ICLR.cc/2024/Conference — ICLR 2024 Conference Withdrawn Submission_

### Official Review · Reviewer_QzZb · 2023-10-31

**Soundness:** 2 fair
**Presentation:** 3 good
**Contribution:** 2 fair
**Rating:** 3
**Confidence:** 4

**Summary:**

In this paper, the authors present a cross-modality learning paradigm for molecular representation. The proposed approach focuses on capturing the correlation between SMILES and molecular graphs through masking and reconstruction techniques. To achieve this, two distinct data modalities are input into separate encoders and subsequently fused within a central backbone model for the reconstruction task.

**Strengths:**

Overall, the motivation behind the proposed method is well-founded. The method's steps are presented clearly and are easily comprehensible, making it accessible for readers to follow and the paper is well-written.

**Weaknesses:**

1. The authors assert the absence of existing work effectively exploring the complementary information between SMILES and graphs, which is contradicted by prior research [1]. This discrepancy could potentially undermine the novelty of the proposed method, and it is crucial for the authors to consider this previous work for comparative analysis. Furthermore, it's important to note that the effectiveness of reconstruction and contrastive learning as pretraining strategies in molecular representation learning is context-dependent. Their efficacy varies based on the specific scenarios they are applied to, and the authors' claim in the last sentence of the fourth paragraph may not universally hold true. Careful consideration and clarification of the conditions under which these strategies are effective are needed for a more precise understanding of their contributions.

2. The two encoders for different modalities in the proposed approach employ different models that have been introduced in prior research. However, the paper falls short in providing a comprehensive explanation regarding the rationale behind their selection and the potential implications if alternative encoders were to be used. Elaborating on the specific reasons for choosing these models and discussing the potential consequences of substituting them with other models is essential to enhance the transparency and completeness of the methodology.

3. Step 2 in the masking strategy appears to be redundant. The authors attempt to mask atoms in SMILES and graphs that do not overlap. The derivations of S2, S3, and S4 also seem unnecessary. It would be helpful if the authors clarify the specific ratios of atoms that are masked in both SMILES and graphs. Additionally, providing information about what replaces the masked atoms is crucial for a complete understanding of the masking strategy.

[1] Zhu, Jinhua, et al. "Dual-view molecule pre-training." arXiv preprint arXiv:2106.10234 (2021).

**Questions:**

1. See the comments above.
2. What is the complementary information between SMILES and molecular graphs the authors are referring to?

---

### Official Review · Reviewer_DRL8 · 2023-11-01

**Soundness:** 3 good
**Presentation:** 3 good
**Contribution:** 2 fair
**Rating:** 3
**Confidence:** 4

**Summary:**

The authors propose a multi-modal framework for learning on molecular data, using SMILES and 2D graph representations. Their framework comprises a text encoder that converts SMILES strings into tokens and a graph encoder that converts molecules into tokens. These tokens are then fed to a joint transformer backbone pre-trained using masked modeling. The masking mechanism masks disjoint regions of the different modalities input to the model.

**Strengths:**

Overall the paper is well presented, and is easy to follow. The authors conduct an extensive ablation and benchmark on many datasets.
It is also a topic well suited to the conferences theme.

**Weaknesses:**

The authors propose to use SMILES and 2d graph representation as two different modalities, though one simply seems to be a "less" informative modality compared to the other, i.e. the 2d graph representation contains what is represented in SMILES and more, hence it is not clear where the gains are coming from.

Although the authors compare many methods, they do not provide comparisons with other transformer based methods that leverage 2d/3d information. Molecules are generally small, and recent STOTA methods are transformers which encode various graph based information in the self-attention mechanism/bias. Further, simply introducing virtual nodes (a token) that is connected to all molecules in the graph results in a dramatic improvement for downstream tasks (as done similarly in graph based methods).

Additionally, the idea of jointly modeling graphs using both SMILES and Graphs has been previously studied, for example [1]. In this work they use SMILES strings and 3D graph structure, with a slightly different motivation where it is quite often the case that downstream datasets do not contain 3D information and one would like to make predictions using the SMILES/2d structural inputs. Providing comparisons/intuition on why the current method improves upon these other methods would strengthen the paper as well.

[1] Cao, Zhonglin, et al. "Moformer: self-supervised transformer model for metal–organic framework property prediction." Journal of the American Chemical Society 145.5 (2023): 2958-2967.

**Questions:**

*see weaknesses

---

### Official Review · Reviewer_82Wx · 2023-11-01

**Soundness:** 3 good
**Presentation:** 3 good
**Contribution:** 3 good
**Rating:** 5
**Confidence:** 4

**Summary:**

The paper introduces a multi-modal pretraining method designed to enhance molecular representation learning. This approach uniquely integrates two modalities: the SMILES string and the molecular graph. To optimize the interplay between these modalities, the author proposes a distinct non-overlapping masking technique, ensuring that no masked tokens in the SMILES string coincide with those in the molecule graph. Extensive experiments are conducted to verify the superiority of the methods on properties prediction benchmarks.

**Strengths:**

1.	This paper utilizes established molecule encoding techniques and merges two distinct molecule modalities for pretraining. The proposed method is simple and effective.
2.	Extensive experiments are conducted to show the effectiveness of the method, as well as ablation studies on the non-overlap mask ratio strategy and the finetuning of the backbone model.

**Weaknesses:**

1. The primary issue lies in the insufficient justification of how SMILES and molecular graphs synergize. While SMILES and graphs are interconvertible, it's recommended for the author to delve deeper, providing a comprehensive explanation or conducting experiments to elucidate how they complement well with each other. This would showcase how they collaboratively offer a superior molecular representation.

2. The technical innovation in the paper seems limited. Both the encoder architectures and the feature aggregation modules seem to draw heavily from prior works. For example, GraSeq[1] had previously explored the integration of graph and sequence within a similar framework.

3. In Table 1, some of the results for the top-performing and the second-best often fall within the standard error range of one another. This weakens the claim that MOLESG is the leading method for these specific tasks. Also, the ablation study reveals that various models have distinct capacities for molecule representation. A strong support is required for showing that the chosen modalities complement each other.

[1]. GraSeq: Graph and Sequence Fusion Learning for Molecular Property Prediction, Guo et.al

**Questions:**

See the above weaknesses.

---

### Official Review · Reviewer_Wr3h · 2023-11-02

**Soundness:** 3 good
**Presentation:** 3 good
**Contribution:** 2 fair
**Rating:** 5
**Confidence:** 4

**Summary:**

This paper introduces a multi-modality molecular pretraining model that attempts to simultaneously encapsulate both SMILES and graph information. The model operates on a unified transformer-based backbone network and is evaluted through molecular property prediction tasks such as MoleculeNet and QM9. While the model demonstrates promising outcomes, there is ambiguity in the empirical settings and the fairness of certain comparisons is questionable. Consequently, the results and assertions put forth require further clarification and substantiation.

**Strengths:**

1. This paper introduces MoleSG, a self-supervised learning framework that aims to fusion SMILES and Graph data, enabling the acquisition of fine-grained interactions and complementary knowledge between the two modalities.
2. The paper proposes a novel non-overlapping masking strategy for both SMILES and graphs, showcasing its superiority across various sub-tasks within MoleculeNet.
3. The paper is well-written and easy to understand.

**Weaknesses:**

1. The main results of MoleculeNet do not appear to be entirely convincing. For instance, when looking at the results for BBBP, it is evident that UniMol's performance exhibits a significant disparity when compared to kANO and MoleSG. This difference can be attributed to the unconventional split method utilized by kANO, known as 'scaffold balance,' as opposed to UniMol's use of a more stringent scaffold-based approach. It raises doubts regarding whether the variation in performance can be attributed to the critical influence of differing split methods. It is also worth noting that the performance of Grover and GEM does not align with the results presented in the UniMol paper. This discrepancy is primarily due to UniMol's reevaluation of Grover using the scaffold split method. To ensure a fair comparison, it is imperative that all baseline results adopt the same split settings, and it is equally important to validate that the split method employed is indeed 'scaffold' and not a variant.
2. This paper proposes a pre-training method to learn the fine-grained interactions between two modalities,  highlighting various limitations associated with single-modality approaches.  However, during the fine-tuning phase, only one modality is utilized, and the fusion backbone is discarded. I question whether the fine-grained complementary knowledge learned by the fusion backbone is effectively applied in the downstream tasks. I am also wander why we cannot use both modality with the fusion backbone in finetuning.  Since MoleSG seems to focus on learning to fusion both modalities, rather than solely enhancing  one modality using another modality. The current fine-tuning setup fails to effectively validate the original motivation behind this research.
3. In Table 3, the observed performance of the 'Ours SMILES' setting significantly lags behind that of the 'Ours graph' setting. Given that SMILES and graph representations are mutually complementary, it raises questions as to why all sub-tasks associated with SMILES consistently underperform in comparison to the graph-based counterparts.
4. Further analysis or visualization is necessary to verify the presence of 'fine-grained interactions' between the SMILES and graph representations during the pretraining phase.
5. There have been previous efforts in the field that aim to integrate SMILES and graph representations, as exemplified by works such as [1] to [4]. As such, the novelty of this study appears to be comparatively limited.
[1] Molecular Joint Representation Learning via Multi-modal Information of SMILES and Graphs
[2] UniMAP: Universal SMILES-Graph Representation Learning
[3] Improving molecular pretraining with complementary featurizations
[4] Dual-view Molecule Pre-training

**Questions:**

as stated in the weakness part.